# Glycan-shielded homodimer structure and dynamical features of the canine distemper virus hemagglutinin relevant for viral entry and efficient vaccination

Hideo Fukuhara[1,2]*†, Kohei Yumoto[1]†, Miyuki Sako[3]†, Mizuho Kajikawa[3], Toyoyuki Ose[1], Mihiro Kawamura[1], Mei Yoda[1], Surui Chen[1], Yuri Ito[1], Shin Takeda[1], Mwila Mwaba[1], Jiaqi Wang[1], Takao Hashiguchi[4], Jun Kamishikiryo[3], Nobuo Maita[5], Chihiro Kitatsuji[1], Makoto Takeda[6], Kimiko Kuroki[1], Katsumi Maenaka[1,2,7,8,9]*

[1]Laboratory of Biomolecular Science and Center for Research and Education on Drug Discovery, Faculty of Pharmaceutical Sciences, Hokkaido University, Sapporo, Japan; [2]Division of Pathogen Structure, Research Center for Zoonosis Control, Hokkaido University, Sapporo, Japan; [3]Medical Institute of Bioregulation, Kyushu University, Fukuoka, Japan; [4]Department of Virology, Faculty of Medicine, Kyushu University, Fukuoka, Japan; [5]Institute for Enzyme Research, University of Tokushima, Tokushima, Japan; [6]Department of Microbiology, Graduate School of Medicine and Faculty of Medicine, The University of Tokyo, Tokyo, Japan; [7]Global Station for Biosurfaces and Drug Discovery, Hokkaido University, Sapporo, Japan; [8]Institute for Vaccine Research and Development (HU-IVReD), Hokkaido University, Sapporo, Japan; [9]Core Research for Evolutional Science and Technology, Japan Science and Technology Agency, Saitama, Japan

*For correspondence:
h-fukuhara@czc.hokudai.ac.jp
(HF);
maenaka@pharm.hokudai.ac.
jp (KM)

†These authors contributed
equally to this work

Competing interest: The authors
declare that no competing
interests exist.

Reviewing Editor: Goran Bajic,
Icahn School of Medicine at
Mount Sinai, United States

**Abstract** Canine distemper virus (CDV) belongs to morbillivirus, including measles virus (MeV) and rinderpest virus, which causes serious immunological and neurological disorders in carnivores, including dogs and rhesus monkeys, as recently reported, but their vaccines are highly effective. The attachment glycoprotein hemagglutinin (CDV-H) at the CDV surface utilizes signaling lymphocyte activation molecule (SLAM) and Nectin-4 (also called poliovirus-receptor-like-4; PVRL4) as entry receptors. Although fusion models have been proposed, the molecular mechanism of morbillivirus fusion entry is poorly understood. Here, we determined the crystal structure of the globular head domain of CDV-H vaccine strain at 3.2 Å resolution, revealing that CDV-H exhibits a highly tilted homodimeric form with a six-bladed β-propeller fold. While the predicted Nectin-4-binding site is well conserved with that of MeV-H, that of SLAM is similar but partially different, which is expected to contribute to host specificity. Five N-linked sugars covered a broad area of the CDV-H surface to expose receptor-binding sites only, supporting the effective production of neutralizing antibodies. These features are common to MeV-H, although the glycosylation sites are completely different. Furthermore, real-time observation using high-speed atomic force microscopy revealed highly mobile features of the CDV-H dimeric head via the connector region. These results suggest that sugar-shielded tilted homodimeric structure and dynamic conformational changes are common characteristics of morbilliviruses and ensure effective fusion entry and vaccination.

## eLife assessment

The manuscript presents **valuable** findings, using **solid** techniques and approaches, that shed additional light into how the canine distemper virus (CDV) hemagglutinin might engage cellular receptors and how that engagement impacts host tropism. The structural data and their analysis were thorough and well-presented. The HS-AFM data, which indicate that homodimers may dissociate into monomers - and thus have significant implications for the model of fusion triggering - are very exciting, but require further validation, perhaps by alternate approaches, to bolster the current molecular model of the CDV fusion triggering.

## Introduction

Canine distemper virus (CDV) infects dogs as well as other carnivores, and causes acute, highly contagious, febrile diseases that affect the respiratory, gastrointestinal, and central nervous systems, along with serious temporary immunodeficiency (*Appel and Summers, 1995*; *An et al., 2008*; *von Messling et al., 2004*). In 2008, the Chinese group reported a natural infection of CDV in rhesus monkeys, indicating that the CDV expanded its host range (*Sun et al., 2010*). CDV is an enveloped virus with a non-segmented, negative-strand RNA genome that is classified in the genus *Morbillivirus* of the family *Paramyxoviridae*, to which the measles virus (MeV) also belongs. CDV has the attachment glycoprotein hemagglutinin (CDV-H) at the virus surface, which utilizes signaling lymphocyte activation molecule (SLAM) and nectin cell adhesion molecule 4 (Nectin-4) as specific entry receptors (*Tatsuo et al., 2001*; *Seki et al., 2003*; *Tatsuo et al., 2000*; *Yanagi et al., 2009*). SLAM, which has two immunoglobulin (Ig)-like domains in the extracellular region, is expressed on thymocytes, activated lymphocytes, mature dendritic cells, macrophages, and platelets, reflecting the tropism and immunosuppressive nature of CDV (*Yanagi et al., 2009*; *Schwartzberg et al., 2009*; *Kiel et al., 2005*). On the other hand, Nectin-4 has three Ig-like domains as an extracellular region and is expressed on epithelial cells, permitting transmission from the basal to the apical side for the spread of viruses (*Takeda et al., 2007*; *Tahara et al., 2008*; *Leonard et al., 2008*; *Mühlebach et al., 2011*; *Noyce et al., 2011*; *Noyce and Richardson, 2012*). The mechanism of viral infection is similar to that of MeV, while CDV tends to cause more serious neurological disruption. Our recent report and other studies have demonstrated that the host species of CDV is partly determined by their SLAMs (*Fukuhara et al., 2019*; *Ohishi et al., 2014*), while the host innate immunity should also be considered (*Takeda et al., 2020*). Notably, some recent CDV strains can infect monkeys, such as rhesus and cynomolgus monkeys, raising the possibility of further expansion of host specificities (*Qiu et al., 2011*). Single amino acid substitutions (R519S, D540G, and P541S) can cause CDV-A75/17 or CYN07dV strains to infect Vero cells expressing human SLAM (*Sakai et al., 2013*; *Bieringer et al., 2013*). Therefore, these results highlight the need for caution regarding the evolution of CDV to acquire the infectious activity to human beings.

The crystal structures of the H protein of MeV (MeV-H) revealed a six-bladed β-propeller folded head domain similar to those of Hemagglutinin/Neuraminidase (HN) proteins of the other paramyxoviruses. On the other hand, the MeV-H forms a disulfide-linked homodimer in unique tilted orientation (*Hashiguchi et al., 2007*). With regard to viral entry events, there exist proposed models in which a conformational change of the H protein induced by SLAM or Nectin-4 binding causes a structural change in the fusion (F) protein at the virus surface, leading to fusion of the virus envelope and the plasma membrane (*Lee et al., 2008*; *Yin et al., 2006*). However, the detailed mechanism of fusion triggered by H proteins remains unclear.

Live attenuated vaccines of CDV developed for more than half a century are highly effective, while there are no officially available vaccines for other carnivores, such as lions, ferrets, and earless seals (*Rouxel et al., 2009*). Similarly, related MeV live vaccines are also effective, and the molecular mechanism for efficient vaccination based on the crystal structures of MeV-H and its complexes has been proposed (*Hashiguchi et al., 2007*; *Colf et al., 2007*; *Hashiguchi et al., 2011*; *Zhang et al., 2013*). MeV-H has sugar shields that cover a large surface, except for the binding sites of human SLAM and Nectin-4. The anti-MeV-H antibodies induced by vaccines tend to target the receptor-binding sites (RBSs), ensuring the efficient production of neutralizing antibodies (*Hashiguchi et al., 2011*; *Zhang et al., 2013*). However, the molecular mechanism underlying the efficient vaccination of CDV, which has different sugar modification sites, remains unclear.

Here, we report the receptor-binding properties and crystal structure of CDV-H from a Kyoto Biken vaccine strain at 3.2 Å resolution. These results indicate that the homodimer structure conserved in MeV-H makes the sites for receptor recognition upward and easily accessible, which is consistent with the recently reported cryo-electron tomography (cryo-EM) structure of the ectodomain of wild-type CDV-H (*Kalbermatter et al., 2023*). The sugar modification sites of CDV-H are not conserved in sequence with those of MeV-H at all. Nevertheless, surprisingly, the glycan shields of CDV-H cover a large surface but expose the RBSs, essentially the same system structurally shared with MeV-H. Furthermore, real-time observation of CDV-H by high-speed atomic force microscopy (HS-AFM) was performed, showing potential of CDV-H homodimer dissociation. These results allowed us to update the model for membrane fusion triggered by receptor binding to induce conformational changes in the fusion protein. Our findings provide insights into the molecular and dynamic events of viruses classified in the genus *Morbillivirus* for cell entry and effective vaccinations.

## Results

### Preparation and Nectin-4-binding characteristics of CDV-H

CDV-H consists of an N-terminal cytoplasmic tail, transmembrane region, stalk region, and C-terminal head domain (*Figure 1A*). Surface plasmon resonance (SPR) analyses in two previous reports showed that the CDV-H protein can bind to dog SLAM with a µM range of $K_d$ (*Fukuhara et al., 2019*; *Khosravi et al., 2016*), but there has been no observation that CDV-H binds to the Nectin-4 ligand at the protein level. Here, we examined the binding activity of CDV-H to human and dog Nectin-4. The soluble head domain of CDV-H (residues 149–604, Kyoto Biken strain) with a 6×Histidine tag at the C-terminal site was produced using HEK293T cells as described in our previous report (*Figure 1A–C*; *Fukuhara et al., 2019*). The ectodomains of human Nectin-4 (residues 1–333, designated as hNectin-4) and dog Nectin-4 (residues 1–332; dNectin-4) were expressed in HEK293T cells and purified by Ni-affinity chromatography, followed by size-exclusion chromatography (*Figure 1A*; *Figure 1—figure supplement 1*). SPR analysis revealed that CDV-H binds to dNectin-4 with low affinity ($K_d$ ~ $10^{-7}$ to $10^{-6}$ M) and fast kinetics (*Figure 1D*; *Table 1*). The binding characteristics of the CDV-H–dNectin-4 interaction were similar to those of CDV-H–SLAM, which is typical for receptor recognition of viral attachment proteins. Furthermore, we examined whether CDV-H could bind to hNectin-4. SPR experiments clearly demonstrated that CDV-H exhibits a sufficient response to hNectin-4 with similar binding kinetics to the CDV-H–dNectin-4 interaction, as shown in *Figure 1D* and *Table 1*.

### Crystal structure of CDV-H

To determine the structure of the vaccine strain CDV-H by X-ray crystallography, the expression plasmid encoding soluble CDV-H head domain was transfected into HEK293S GnTI⁻ cells, which lack *N*-acetyl-glucosaminyltransferase I (GnTI) activity to modify glycosylation with homologous oligomannose-type sugars (Man$_5$GlcNAc$_2$) and was purified as described above. The molecular weight of the non-reducing soluble CDV-H determined by gel filtration chromatography and sodium dodecyl sulfate–polyacrylamide gel electrophoresis (SDS–PAGE) matched its dimer form (*Figure 1B*), indicating that the head domain of CDV-H exists as a disulfide-linked homodimer. The crystals were obtained by sitting-drop vapor diffusion at 20°C in 100 mM tri-sodium citrate (pH 5.6), 10% polyethylene glycol-4000, and 10% isopropanol (*Figure 2A*). We determined the crystal structure of CDV-H refined at 3.2 Å resolution by molecular replacement using the MeV-H structure (*Hashiguchi et al., 2007*), which shares approximately 30% sequence identity in the head domain with CDV-H as a search probe (*Figure 2B* and *Table 2*).

There are two CDV-H monomers (chains A and B) in the asymmetric unit of the crystals, which have only slight differences (root mean square deviation = 0.301 Å) in the structures of some loops (*Figure 2—figure supplement 1*). Therefore, hereafter, we focus on chain A as a representative structure of CDV-H, in which a more extensive electron density is observed than that in chain B. The head domains of CDV-H of vaccine strain exhibit a six-bladed β-propeller structure, which is essentially the same as recent cryo-EM structure of wild-type CDV-H ectodomain (r.m.s.d. = 0.584–0.743 Å) (*Kalbermatter et al., 2023*), similar to that of the MeV-H structures (*Figure 2C*). Furthermore, CDV-H forms a homodimer structure facing the β-propeller 2, α1 and α2 helices, resulting in a dimer orientation inclined to the horizontal plane (*Figure 3A*), similar to the MeV-H homodimer structures and distinct

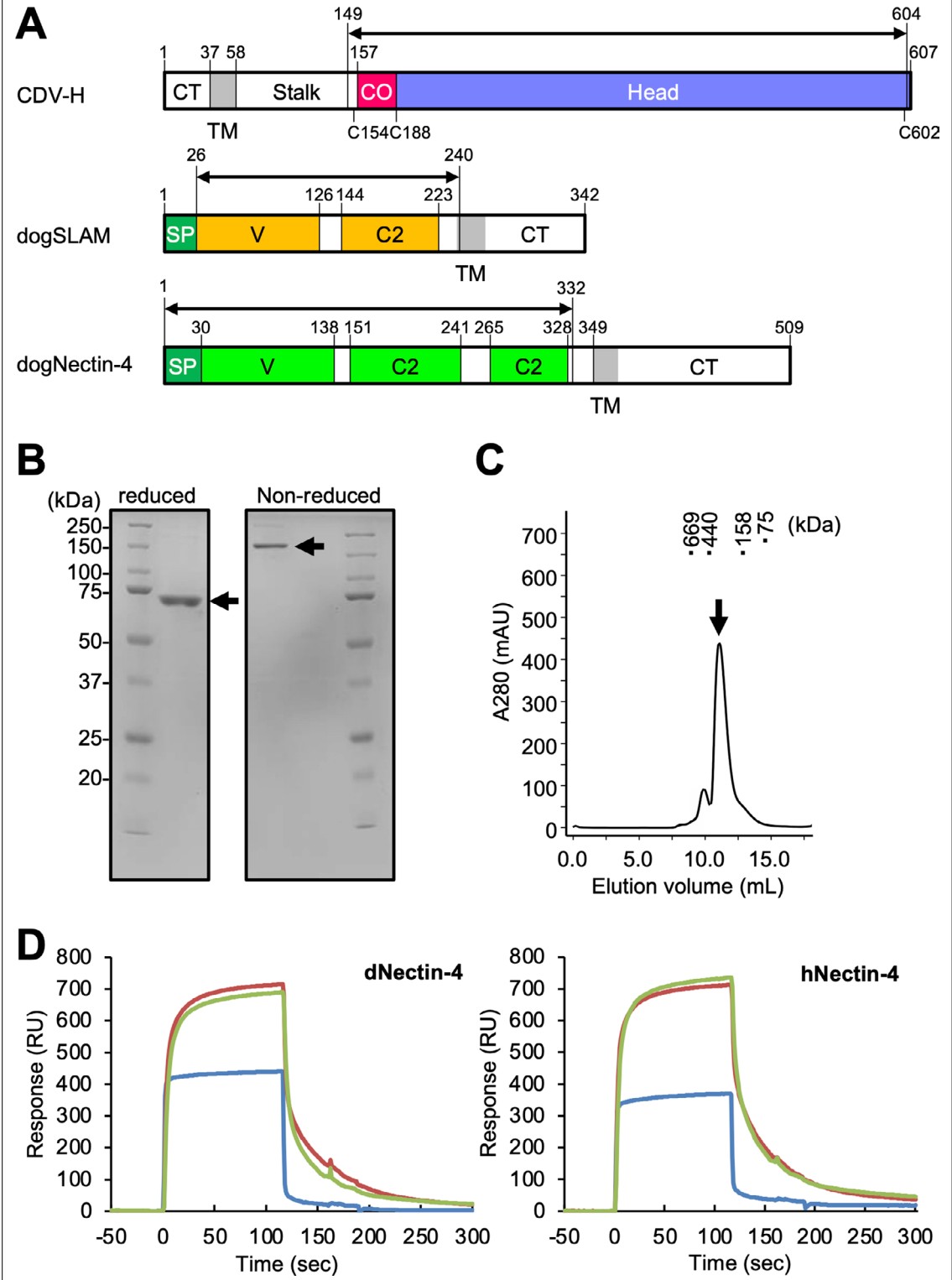

**Figure 1.** Protein preparation and Nectin-4 binding to CDV-H. (**A**) Schematic images for expression region of CDV-H globular domain (top), dog signaling lymphocyte activation molecule (SLAM) ectodomain (middle), and dog Nectin-4 (bottom). CT, cytoplasmic tail; TM, transmembrane; CO, connector region; SP, signal peptide. (**B**) Sodium dodecyl sulfate–polyacrylamide gel electrophoresis (SDS–PAGE) for CDV-H protein in reducing (left) and non-reducing (right) conditions. The arrows indicate monomer (left) and disulfide-linked dimer (right) of CDV-H. (**C**) Gel filtration chromatogram of CDV-H using Superdex 200 column. The arrow indicates the peak of CDV-H (Ve = v12.2 ml). (**D**) Surface plasmon resonance analysis for the interactions

*Figure 1 continued on next page*

*Figure 1 continued*

between CDV-H and Nectin-4 receptors: dog (left) and human (right) Nectin-4. The green, red, and blue lines indicate CDV-Hwt, CDV-Hvac, and β2 microglobulin as a negative control, respectively.

The online version of this article includes the following figure supplement(s) for figure 1:

**Figure supplement 1.** Purification of dog and human soluble Nectin-4.

from other paramyxoviruses, such as parainfluenza virus 5 (*Figure 3B*; *Hashiguchi et al., 2007*; *Yin et al., 2006*; *Yuan et al., 2011*). The interactions at the homodimer interface are mainly composed of three parts: the top, middle, and bottom sections, as shown in *Figure 3A*. The top section of CDV-H is a hydrophobic cluster consisting of Leu201, Val203, Trp259, Leu260, and Leu265, which interact with Leu205, Leu209, Leu234, and Ile248, respectively. Trp259 is substituted for Pro263 in MeV-H, compensated by the replacement of Leu209 with Tyr209, resulting in the maintained CH–π interaction between Leu209 and Trp259 (*Figure 3C*, top). The middle section is a hydrogen-bond network formed by the main-chain O atoms of Thr268 and Thr269, with the Nδ atom of Gln267 (*Figure 3C*, middle). Lys257 forms a hydrogen bond with the side-chain O atom of Asn270. Tyr271 interacts hydrophobically with Lys257, Leu265, and Gln267. Tyr271 further interacts with Trp259 at the interface between the top and middle sections. In MeV-H, His271 forms a hydrogen bond with the side-chain O atom of Met272 in another monomer. In the bottom section of CDV-H, Phe323 of each molecule is hooked by a π–π interaction (*Figure 3C*, bottom). Phe323 further interacts with the hydrophobic residues, Phe255, Ile322, and Pro169. In MeV-H, Trp327 makes π–cation interactions with Lys328 of another monomer. The characteristics of these interactions at the homodimer interface are conserved in morbilliviruses, while some HN proteins in the paramyxoviruses have substitutions and lack middle or bottom sections, adopting different angles of homodimer formation (*Figure 3B*). These findings suggest that the unique dimer orientation is a common feature of morbilliviruses and is appropriate for binding to the protein receptors SLAM and Nectin-4 (*Figure 3—figure supplement 1*).

## Receptor-binding sites

The previously reported complex structures of MeV-H with SLAM or Nectin-4 revealed that the β4–β6 blades are involved in their interactions (*Hashiguchi et al., 2011*; *Zhang et al., 2013*; *Santiago et al., 2010*). The superposition of the current CDV-H structure onto each individual complex structure enabled the construction of a reasonable CDV-H–receptor complex model without any significant structural hindrance (*Figure 4—figure supplement 1*). Previous mutagenesis studies have reported that eight amino acids (Tyr525, Asp526, Ile527, Ser528, Arg529, Tyr547, Thr548, and Arg552) on the β5 blade of CDV-H are important for interaction with dSLAM (*von Messling et al., 2005*; *Zipperle et al., 2010*). The crystal structure of MeV-H complexed with tamarin SLAM showed that the H-SLAM interaction composes mainly four sites (*Figure 4A*; *Hashiguchi et al., 2011*). At the corresponding site 1 on CDV-H, Asp501 is disordered in both asymmetric units of CDV-H. Asp501 and Asp503 on the acidic loop of the β5 blade are equivalent to Asp505 and Asp507 in MeV-H interacting with Lys77 and Arg90 of tamarin SLAM (*Hashiguchi et al., 2011*). Both Lys77 and Arg90 are conserved in dSLAM. Next to the acidic loop, Arg529 of the β5 blade supported by Asp526 on CDV-H would interact with

**Table 1.** Kinetics of Nectin-4 against MeV-H and CDV-H.

**SPR results**

| | dNectin-4 | | hNectin-4 | |
|---|---|---|---|---|
| | $K_A$ [M$^{-1}$] | $K_D$ [M] | $K_A$ [M$^{-1}$] | $K_D$ [M] |
| MeV-Hwt | $7.39 \times 10^5$ | $1.35 \times 10^{-6}$ | $1.48 \times 10^6$ | $6.74 \times 10^{-7}$ |
| MeV-Hvac | $9.50 \times 10^5$ | $1.05 \times 10^{-6}$ | $1.50 \times 10^6$ | $6.66 \times 10^{-7}$ |
| CDV-Hwt | $3.96 \times 10^5$ | $2.53 \times 10^{-6}$ | $3.97 \times 10^5$ | $9.48 \times 10^{-7}$ |
| CDV-Hvac | $1.06 \times 10^6$ | $9.42 \times 10^{-7}$ | $1.05 \times 10^6$ | $2.52 \times 10^{-6}$ |

The values of dissociation and association constants ($K_D$ and $K_A$) were determined by SPR analysis.

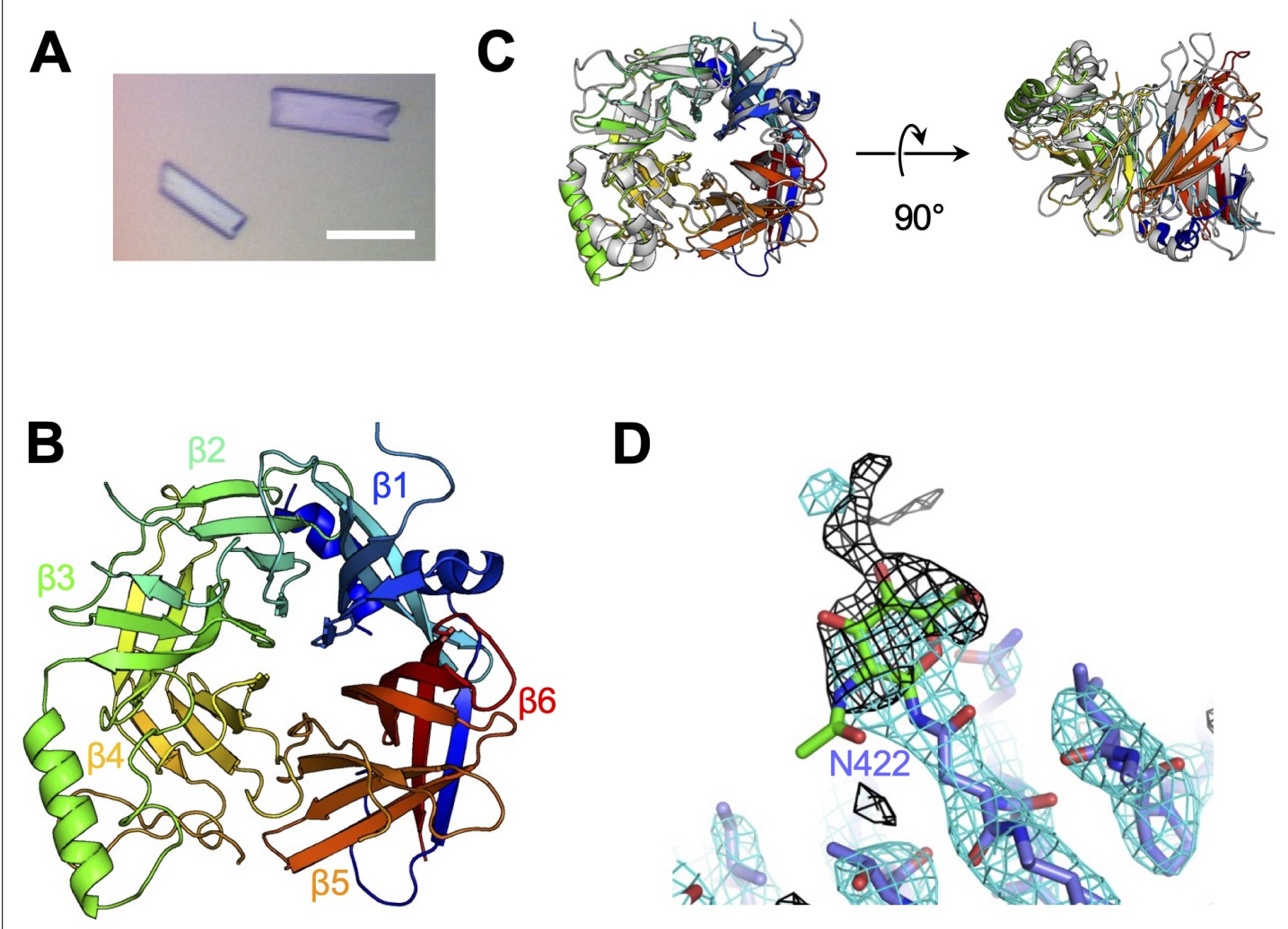

**Figure 2.** Crystal structure of CDV-H. (**A**) Micrograph of CDV-H head domain crystal. Scale bar indicates 0.1 mm. (**B**) Overall structure of CDV-H with secondary structures in cartoon model, colored gradually from blue (N-terminus) to red (C-terminus). (**C**) Structural comparison of CDV-H (rainbow color) and MeV-H (PDB ID: 2ZB6, gray). (**D**) Electron density map (2.5 σ omit map shown in black mesh) around Asn422 of CDV-H attached with *N*-glycan.

The online version of this article includes the following figure supplement(s) for figure 2:

**Figure supplement 1.** Electron density maps of CDV-H chains A and B.

Glu123 of dSLAM in a manner similar to site 2 of the MeV-H–tamarin SLAM complex (*Figure 4B, C*). These interactions are likely strengthened by the surrounding hydrophobic interactions formed by Pro554 and Phe552 on MeV-H and His61 on tamarin SLAM, which correspond to residues Pro550 and Thr548 on CDV-H and His61 on dSLAM, respectively. Additionally, Leu63 of dSLAM, equivalent to Val63 of tamrin SLAM, is likely to be involved in hydrophobic interactions around site 2. Consistently, Ohno also highlighted the importance of Leu63 in dSLAM for CDV infection (*Ohno et al., 2003*). While site 3 residues on the CDV-H edge of β-propeller 6 have a low conservation rate, this region interacts with β8 of tamarin SLAM in the complex structure of MeV-H–tamarin SLAM. Furthermore, the β8-H contact-forming residues on dSLAM are conserved, except for His130. Therefore, this region can interact with dSLAM without side-chain dependency. At site 4, the aromatic residues Tyr520, Tyr537, Tyr539, and Thr548 on CDV-H are conserved, except for the substitution of Phe552 with Thr548. Additionally, Ile74, of dSLAM equivalent to Val74, tamarin SLAM is stabilized by hydrophobic Leu464, Phe483, and Leu500 on CDV-H. Sequence comparison between dSLAM and tamarin SLAM showed that 7 of 21 residues at the H-binding face (Pro70, Gly71, Ile74, Lys75, Lys76, Phe126, and His130 in dSLAM) are different (*Figure 4C*). The amino acids corresponding to residues 74, 75, 76, 126, and 130

**Table 2.** Data collection and refinement statistics.

| | CDV-H |
|---|---|
| **Data collection** | |
| Space group | $P4_32_12$ |
| Cell dimensions | |
| $a = b, c$ (Å) | 86.84, 303.51 |
| Resolution (Å)* | 34.8–3.1 (3.2–3.1) |
| $R_{merge}$ | 0.153 (0.914) |
| $I/\sigma I$ | 11.2 (3.0) |
| Wilson $B$-factor | 69.86 |
| CC1/2 | −0.808 |
| Completeness (%) | 99.6 (100) |
| Redundancy | 6.1 (6.3) |
| | |
| **Refinement** | |
| Resolution (Å) | 34.8–3.1 (3.2–3.1) |
| No. reflections | 21,519 (2131) |
| $R_{work}/R_{free}$ | 0.2645 (0.3532)/0.3092 (0.4064) |
| No. atoms | |
| Protein | 6564 |
| $B$-factors | |
| Protein | 70.56 |
| R.m.s. deviations | |
| Bond lengths (Å) | 0.003 |
| Bond angles (°) | 0.72 |
| Ramachandran plot[†] | |
| Favored | 94% |
| Allowed | 6% |
| Outliers | 0.73% |

*Highest resolution shell is shown in parenthesis.
[†]Analyzed by Molprobity (**Harder et al., 1996**).

of dSLAM reportedly influence the host specificities of morbilliviruses (*Ohishi et al., 2010*). Therefore, CDV-H binds to dSLAM in a manner similar to MeV-H–tamarin SLAM interaction.

Next, the crystal structure of MeV-H in complex with hNectin-4 showed that the H–hNectin-4 interaction consists of three main sites (*Figure 5A*; *Zhang et al., 2013*). To examine CDV-H binding to dog Nectin-4, the superposition of CDV-H onto the MeV-H–hNectin-4 complex structure (*Zhang et al., 2013*) revealed that 10 of the 24 interacting residues were identical in MeV-H with CDV-H, while all 20 interacting residues of hNectin-4 were identical to dNectin-4 (*Figure 5B, C*). As mentioned above, SPR analysis demonstrated that both dNectin-4 and hNectin-4 could bind to CDV-H (*Figure 1D*). This result suggests that the binding mode of CDV-H to dNectin-4 is presumably the same as that of MeV-H to hNectin-4. Zhang et al. defined the interaction area as sites I–III in the complex structure of MeV-H with hNectin-4 (*Zhang et al., 2013*). Along with the definition, the site I includes hydrophobic groove composed of eight residues, which are maintained; the Leu464, Leu482, Phe483, and Leu526

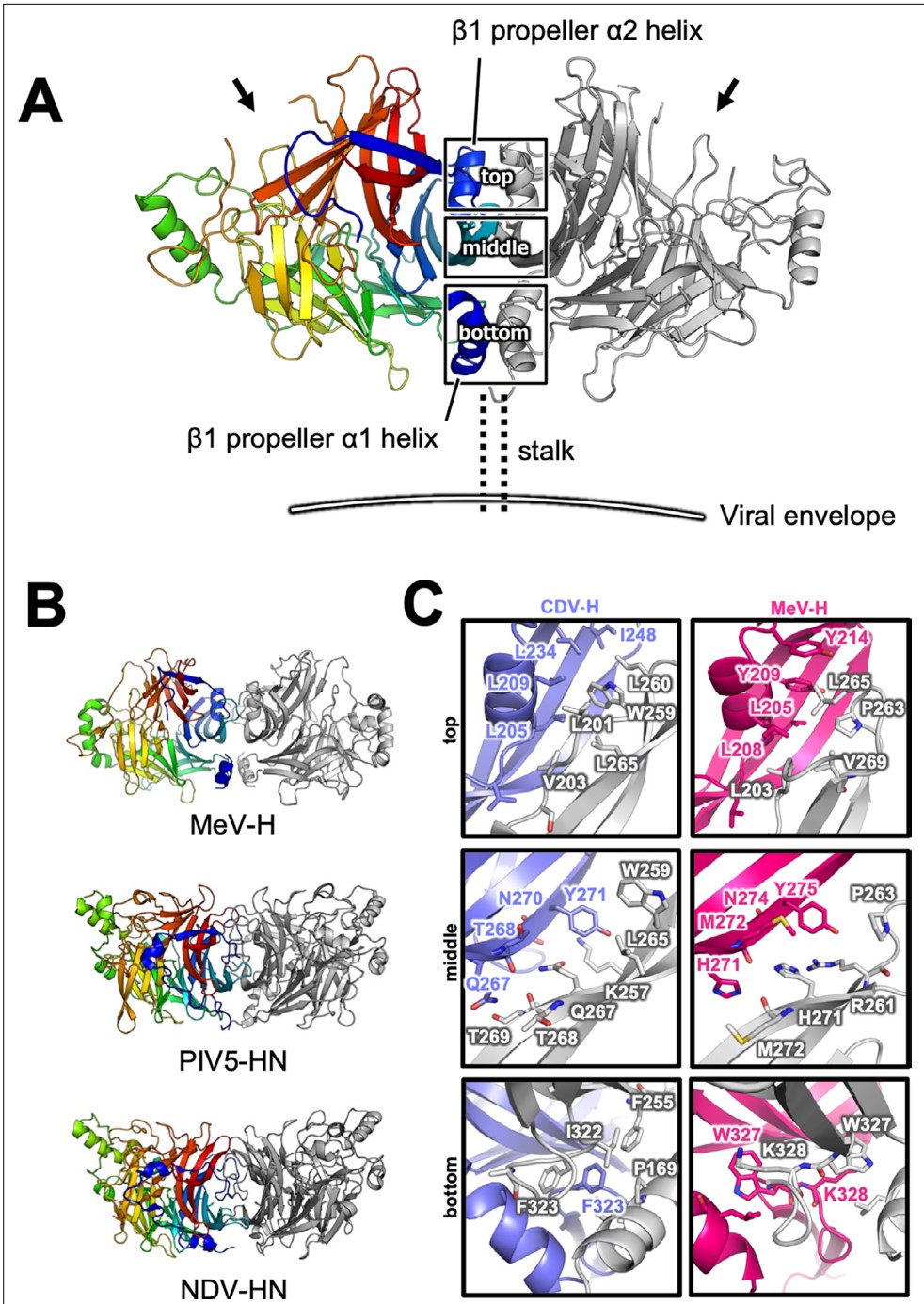

**Figure 3.** Homodimer structure and its interface of CDV-H. (**A**) Homodimer structure of CDV-H with secondary structures in cartoon model as the same as *Figure 2B* (rainbow: chain A and gray: chain B). The black arrows indicate the receptor-binding sites. (**B**) Structural comparison of CDV-H with other paramyxovirus hemagglutinin dimers. MeV-H (top, PDB code: 2ZB6), PIV5-HN (middle, PDB code: 4JF7), and NDV-HN (bottom, PDB code: 3T1E) are shown in similar direction with CDV-H of **A**. (**C**) The closed-up images for three sites at the homodimer interface. One of the CDV-H dimer is shown in slate, asymmetric unit in gray. One of the MeV-H dimer is colored hot pink.

The online version of this article includes the following figure supplement(s) for figure 3:

**Figure supplement 1.** Structure-based alignment of receptor-binding glycoproteins from various paramyxoviruses.

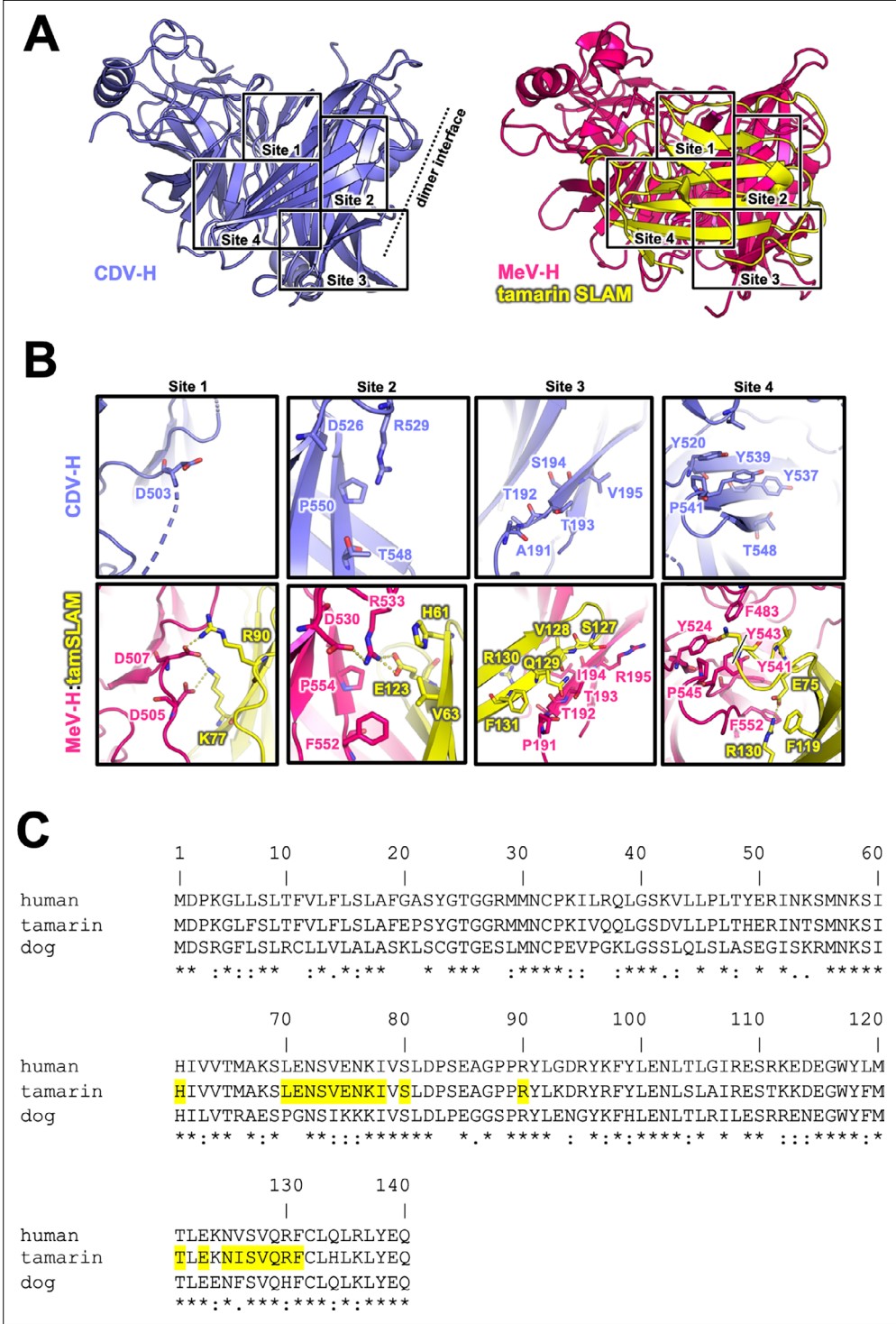

**Figure 4.** Putative signaling lymphocyte activation molecule (SLAM)-binding site on CDV-H. (**A**) Structures of CDV-H with putative SLAM-binding sites (left, slate) and SLAM complex of MeV-H (right, hot pink) are shown. (**B**) The residues that are involved in SLAM receptor binding revealed by mutagenesis studies are shown in stick model on CDV-H (top) and MeV-H (bottom). (**C**) Amino acid sequence alignment of v-type domains of SLAMs. Asterisks (*) indicate fully conserved residues. A colon (:) indicates conservation between groups of strongly similar properties. A period (.) indicates conservation between groups of weakly similar properties. Interacting with hemagglutinin residues (<4.0 Å) are indicated in yellow-colored boxes.

The online version of this article includes the following figure supplement(s) for figure 4:

**Figure supplement 1.** The superposition of CDV-H onto the MeV-H complex structure.

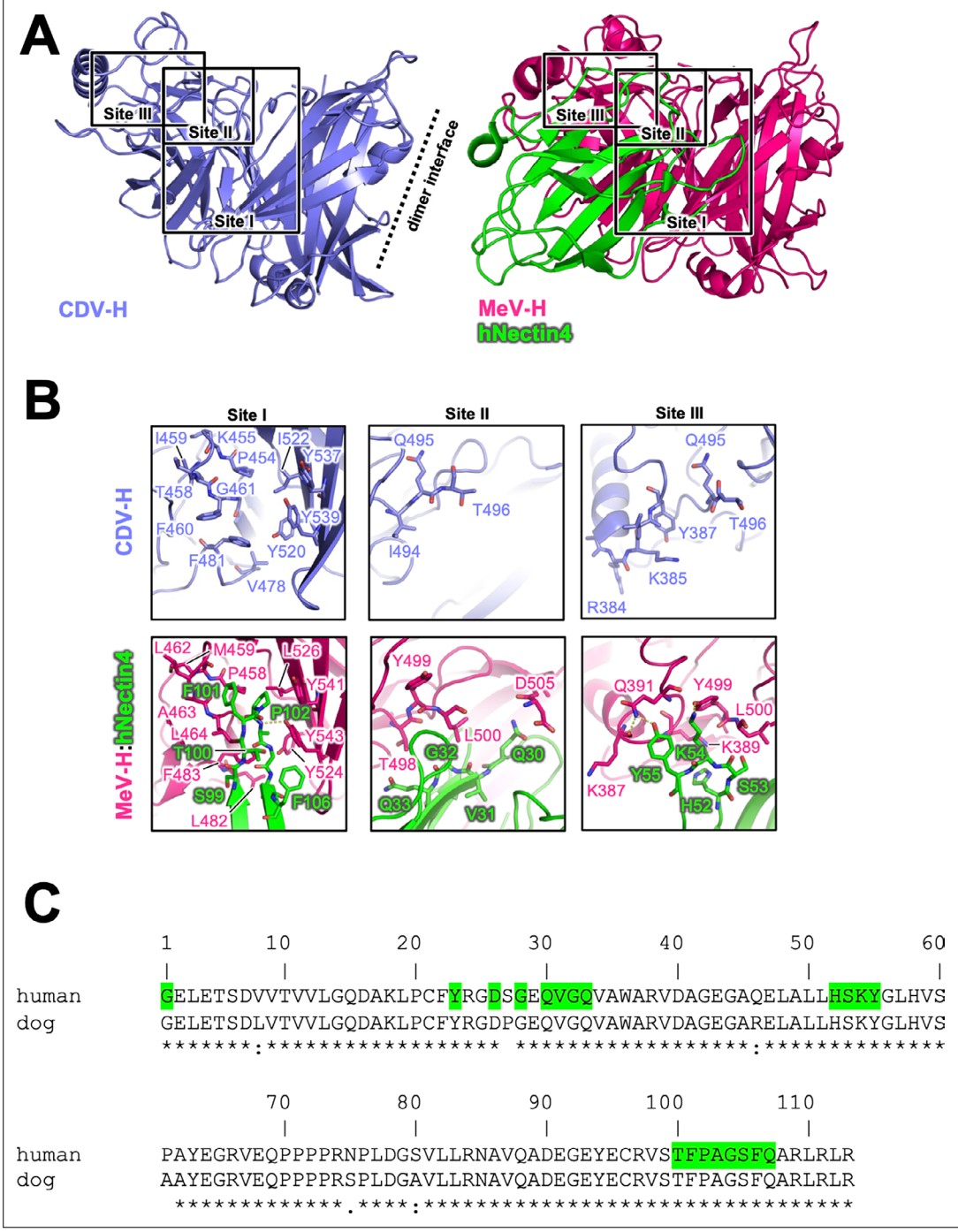

**Figure 5.** Putative Nectin-4-binding site on CDV-H. (**A**) Structures of CDV-H with putative Nectin-4-binding sites (left, slate) and Nectin-4 complex of MeV-H (right, hot pink) are shown. (**B**) The residues that are involved in Nectin-4 receptor binding revealed by mutagenesis studies are shown in stick model on CDV-H (top) and MeV-H (bottom). (**C**) Amino acid sequence alignment of v-type domains of Nectin-4s. Abbreviations and green-colored boxes are shown in the same manner with *Figure 4C*.

of MeV-H are substituted to Phe460, Val478, Leu479, and Ile522 in CDV-H, respectively (*Figure 5B*). The change from Leu500 of MeV-H to Thr496 suggests a lower contribution of site II, as supported by a previous report (*Langedijk et al., 2011*). At site III, Gln391 and Tyr499 of MeV-H are substituted by Tyr387 and Gln495 of CDV-H, which interact with Tyr55 and Lys54 of Nectin-4 the receptors, respectively.

## Glycosylation sites

Previous studies have suggested that the H proteins of a wild-type strain isolated from cases of natural CDV infections in recent years and vaccine strains show approximately 90% identity in their open reading frame, but may have differences in antigenicity of viruses because the wild-type strain has more potential *N*-linked glycosylation sites in its H protein (*Harder et al., 1996*; *Iwatsuki et al., 1997*; *Lan et al., 2006*; *Haas et al., 1997*). There were five potential glycosylation sites (Asn149, Asn391, Asn422, Asn456, and Asn587) in the H protein head domain of the Kyoto Biken vaccine strain (CDV-Hvac) and two additional sites (Asn309 and Asn603) in the 5VD wild-type strain (CDV-Hwt) (*Figure 6A*). To identify the *N*-glycosylation sites, mutagenesis was performed on each site of CDV-Hs. Western blot analysis indicated that all asparagine CDV-H mutants except for the N309D and N603D mutants migrated further than the wild-type, revealing that both CDV-Hvac and CDV-H wt have five sugar chains in common glycosylation sites (Asn149, Asn391, Asn422, Asn456, and Asn587) (*Figure 6B*), while Asn309 and Asn603 specific to CDV-Hwt are not glycosylated. This result is consistent with a previous report showing that five standard glycosylation sites exist in the ecto-domain of the wild-type CDV-H protein (*Sawatsky and von Messling, 2010*). The CDV-H structure, which contains five *N*-linked sugars as described above, exhibits electron density only for Asn422-linked GlcNAc (*Figure 2D*). Asn149- and Asn422-linked sugars, which are located close to each other, appear to bury internal spaces between the stalk and head domains, and Asn587-linked sugars might support the stabilization of the CDV-H dimer orientation (*Figure 6C*, *Figure 6—figure supplement 1*). Sugars modified on Asn391, Asn456, and Asn587 seemed to cover the large pocket in CDV-H, and Asn456-linked sugars appeared to lean toward the center of the pocket (*Figure 6C*). Finally, since the receptors, SLAM and Nectin-4, bind to the side surface of CDV-H conserved with MeV-H, as revealed by the abovementioned mutagenesis study, the majority of CDV-H surfaces are covered with sugars, forcing only the RBSs to be exposed (*Figure 6C*).

## Mobile interdomain features of CDV-H dimer revealed by the HS-AFM

The crystal structure of CDV-H shows a connector domain (residues 158–187) that mediates a head domain to the stalk region, which facilitates head-to-head homodimer formation with a set of disulfide bonds at Cys154 (*Figures 1A and 7A*). This region was mainly disordered in the previous MeV-H structures (*Hashiguchi et al., 2007*). Interestingly, there were no covalent bonds in the homodimer interface of the head domains, even though a conserved homodimer orientation was observed in the crystal structures of CDV-H and MeV-H, as described above. This observation raises the possibility that each head domain of CDV-H also dissociates and moves flexibly, as shown in the structure of Nipah virus (NiV)-G protein, previously (*Wang et al., 2022*). Therefore, here, HS-AFM was used to directly visualize the real-time dynamics of the CDV-H under physiological conditions. When CDV-H was loaded onto a mica substrate and scanned with a cantilever to acquire images of attached molecules, the CDV-H dimer was observed as two globules clustered together in most cases, but sometimes, each domain moved independently (*Figure 7B* and *Videos 1–3*). Time-course analysis of the dynamics of the representative CDV-H dimer showed that CDV-H could adopt both associated and dissociated forms (*Figure 7C*). The distances between the domains were calculated by measuring those between the centers of mass of each domain. Finally, the distribution of distances between each head domain in the CDV-H dimers showed approximately 15 nm as a major peak (*Figure 7D*). This is a reasonable length for the linker between the head domain dimers.

## Discussion

In this study, we determined the crystal structure of the vaccine strain CDV-H head domain with a connector domain. The overall structure of the head domain shows a folding similar to that of MeV-H (*Hashiguchi et al., 2007*) and a homodimer with a tilted domain orientation is formed, essentially the same as the MeV-H homodimer, which is consistent with the cryo-EM structure of wild-type CDV-H ectodomain (*Kalbermatter et al., 2023*). The binding of CDV-H to dNectin-4 was demonstrated by SPR, revealing low-affinity and fast-kinetic interactions, similar to SLAM binding. The conventional idea that low-affinity and fast-kinetics interactions between viral attachment proteins and their receptors on target cells do not cause significant structural changes (*Dimitrov, 2004*), suggesting that CDV-H-receptor recognition is essentially similar to that of MeV.

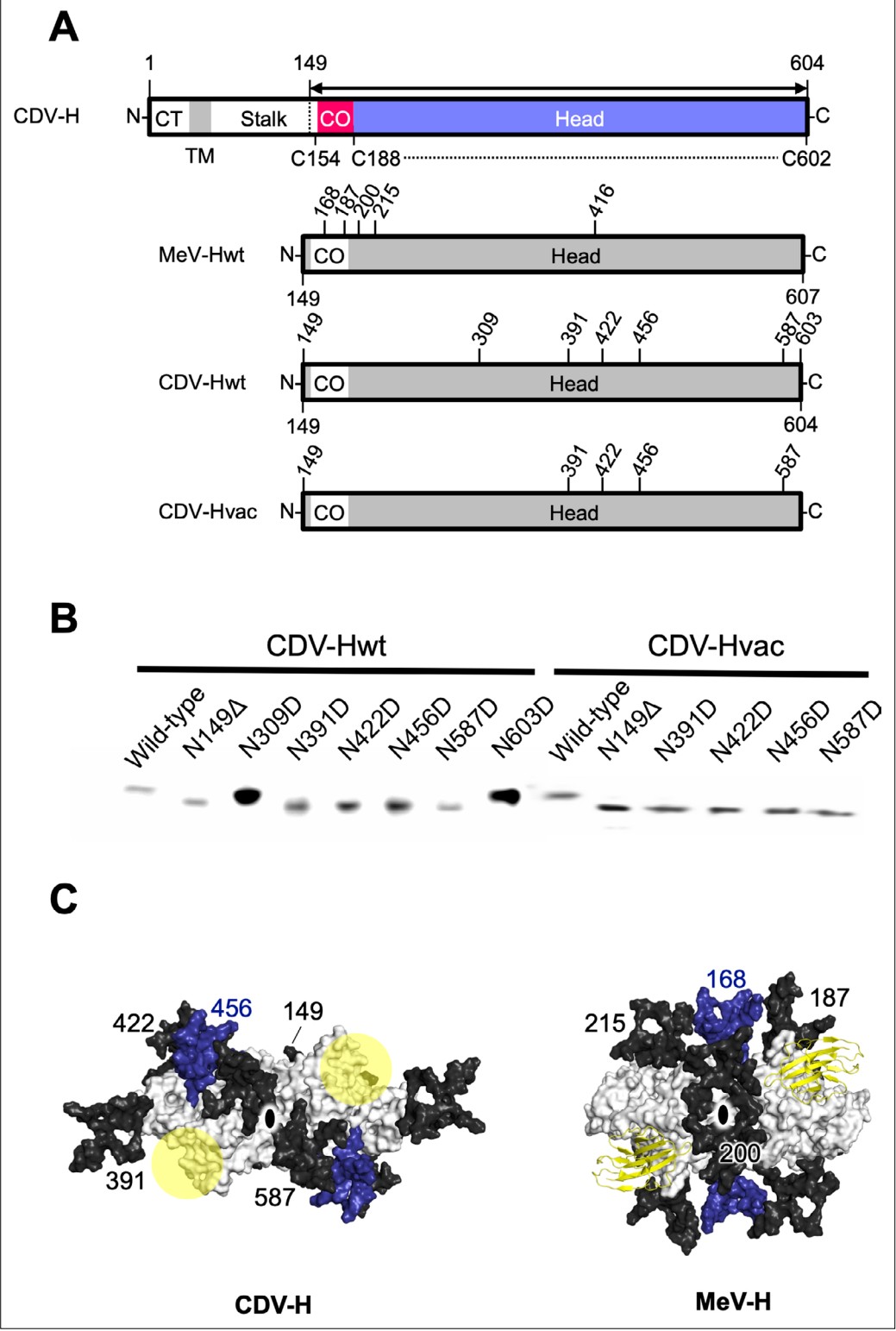

**Figure 6.** Glycan shield of CDV-H. (**A**) Schematic images of full-length and expression constructs of ectodomain of CDV-Hs (wild-type and vaccine strains), together with that of MeV-H. (**B**) Mutagenesis studies for all potential glycan modification revealed to show that N390D and N630D showed clear western blot data. (**C**) Differences in glycan shields. Glycosylation sites on the surface of CDV-H (left) and MeV-H (right) dimers shown in white, with common *N*-linked glycans modeled in black and deep blue. Receptor-binding sites were shown in yellow circle on

*Figure 6 continued on next page*

*Figure 6 continued*

the CDV-H dimer. Tamarin signaling lymphocyte activation molecules (SLAMs) bound to the MeV-H dimer were shown in yellow. Twofold symmetry axis was shown as black ellipse.

The online version of this article includes the following figure supplement(s) for figure 6:

**Figure supplement 1.** Glycan modifications onto CDV-H and MeV-H–signaling lymphocyte activation molecule (SLAM) complex.

Our previous report and other studies have demonstrated that the host species of CDV are restricted by differences in SLAM between animals (*Fukuhara et al., 2019*; *Ohishi et al., 2014*). SLAM-interacting residues in CDV-H, predicted by superimposing the MeV-H–SLAM complex structure onto the CDV-H structure, are conserved. Notably, the mutations T496L and T548F in CDV-H were not sufficient to switch the SLAM receptor preference (*Fukuhara et al., 2019*). Furthermore, a previous report demonstrated that the N-terminal region (position 29) of hSLAM contributes to binding to MeV but not to human-adapted CDV (*Seki et al., 2020*). These findings suggest that SLAM recognition sites are similar but somehow different between CDV and MeV. In contrast, interacting residues of hNectin-4 with MeV-H in the crystal structure of their complexes were completely conserved in dNectin-4 (*Figure 5C*). In accordance with this result and crystal structures, the SPR study showed that CDV-H binds to hNectin-4 with low affinity (~µM range) and fast kinetics, comparable to that of dNectin-4, as described above (*Figure 1D*). Therefore, in contrast to SLAM binding, Nectin-4 recognition was highly conserved in MeV- and CDV-H. These results raise the possibility that future CDV-H variants acquiring human SLAM binding may have the potential to expand host specificity to infect human beings, and the cross-reactivity of antibodies targeting the RBSs of CDV and MeV is expected.

While vaccines for CDV are quite effective and widely used, some groups insisted that the number of *N*-linked glycosylation sites in the wild-type CDV-H increased to weaken the immune responses induced by CDV vaccines (*Harder et al., 1996*; *Iwatsuki et al., 1997*). However, a previous study together with this work clarified that the additional *N*-linked glycosylation sites, Asn309 and Asn603, of CDV-H are not modified with sugars (*Figure 6B*; *Sawatsky and von Messling, 2010*). Thus, the number of sugars in the wild-type CDV-H was presumably maintained, ensuring that the CDV vaccine strains developed in the 1960s have been still effective. Additional passages could lose two *N*-glycosylation sites (Asn391 and Asn456) to produce CDV-H harboring three *N*-linked sugars (*Haig, 1948*). The three sugars linked to Asn149, Asn422, and Asn587 were considered to be fundamental for the correct folding of CDV-H. However, the glycosylation sites mentioned above are not conserved in MeV-H (*Figure 6A*). Surprisingly, however, we mapped glycans onto the CDV-H structure, showing that the glycan shield was established, except for putative RBSs, similar to MeV-H (*Figure 6C*). Our findings provide structural and functional insights into the mechanism of the highly efficient induction of neutralizing antibodies, which are common to MeV-H, even though different sugar modifications (*Hashiguchi et al., 2007*), contribute to the development of effective vaccines against other viruses.

Two glycoproteins, H and F, coordinate to achieve membrane fusion during viral entry as follows. The interaction between the H protein and the receptor induces activation of the F protein by H–F interaction. Subsequent irreversible refolding of the F protein causes its fusion peptide and transmembrane region to be closely located to form the fused pore. There are several kinds of the viral entry mechanisms for morbilliviruses proposed, such as the 'safety-catch', 'bi-dentate', and 'oligomerization' models (*Fukuhara et al., 2020*; *Avila et al., 2015*; *Liu et al., 2013*; *Gui et al., 2015*). The crystal structural analyses of MeV-H and MeV-H–SLAM complex (*Hashiguchi et al., 2011*) revealed two kinds of orientations (forms I and II) of the dimer of the MeV-H dimers, prompting us to propose a viral entry model in which form I (many interface areas between the dimers) can transform to form II (fewer interfaces of the dimers but stabilized upon SLAM complex formation). Considering further evaluation of the models, we recently proposed a possible model (*Fukuhara et al., 2020*) as follows: (1) the stalk region of the H protein binds to the putative H protein-binding site of F protein on the viral surface, (2) the H protein interacts with the host receptors at the virus–cell interface to facilitate the orientation shift of dimers of the H homodimers, and (3) the H protein binds to an additional site of the F protein and/or opens the space to induce the refolding of F protein in pre- to post-fusion states. In this study, the crystal structure of CDV-H included a homodimer with unique intermolecular angles, common to MeV-H, exposing the presumed RBS to the top side far from the viral membrane, which is likely beneficial for access to receptor binding. While the homodimer angles of CDV-H and

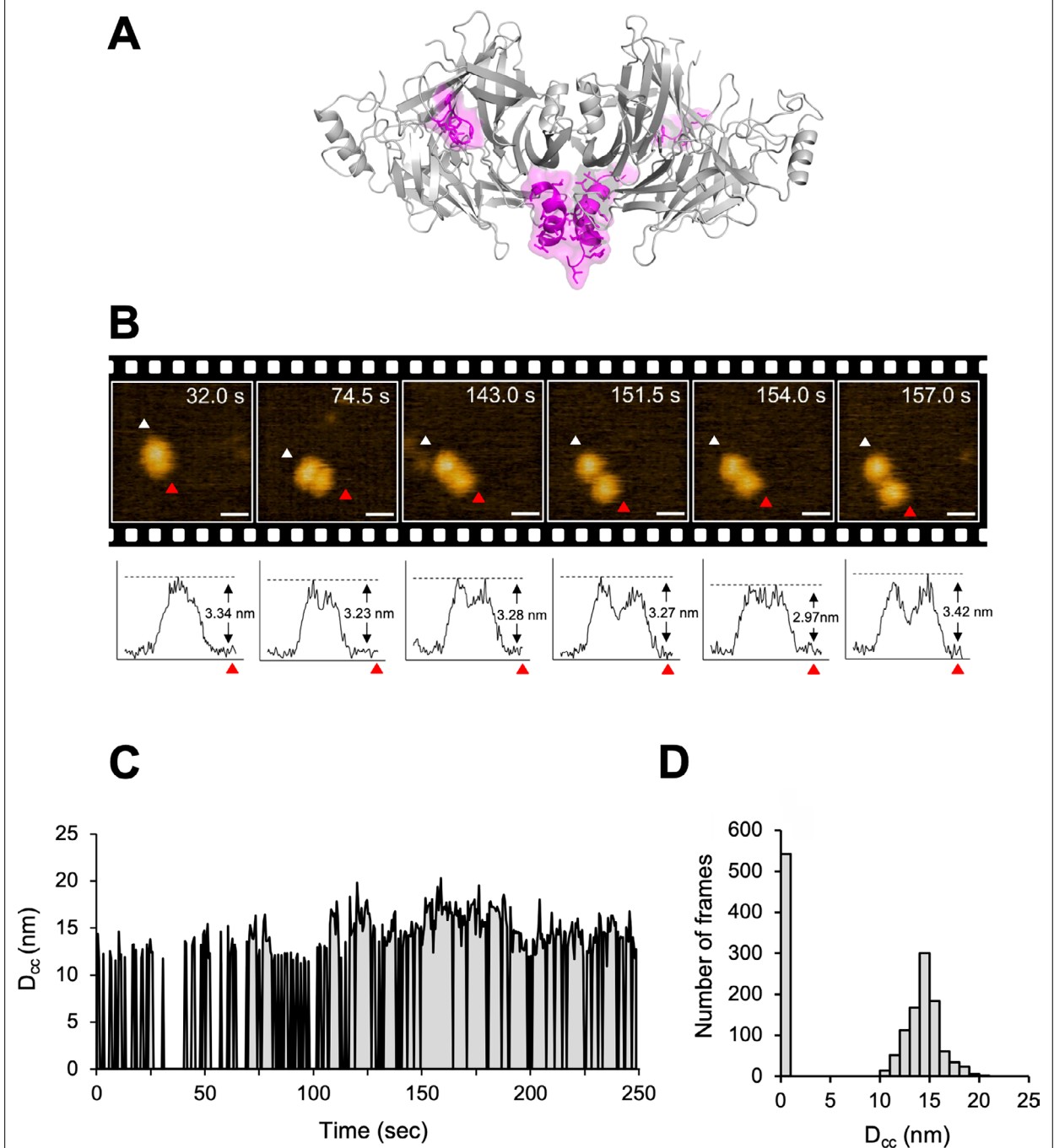

**Figure 7.** Structural dynamics of CDV-H analyzed by high-speed atomic force microscopy (HS-AFM). (**A**) Homodimeric structure of CDV-H with a part of stalk region (residues 149–604). The globular head domain (gray) and the stalk region (magenta) are shown in cartoon model. (**B**) Serial HS-AFM images of CDV-H homodimer at 2 fps are shown. Scale bar indicates 20 nm length. Graphs under each HS-AFM image indicate the z-axis parameter between red and white triangle. (**C**) The distances between signal centers ($D_{cc}$) of representative homodimer were monitored at time scales. (**D**) The numbers of CDV-H homodimers are classified as several distances.

MeV-H are different from those of other paramyxoviruses, such as parainfluenza virus (PIV5), the RBSs are all exposed to the top site distal to the viral membrane (*Figure 3A, B*). The homodimer interface and flexible linkage between each head domain are conserved, suggesting that this structural feature of homodimers is relevant for the fusion step of viral entry. Next, the direct and real-time observation of viral glycoprotein structure using HS-AFM revealed that the homodimer of the CDV-H head domains showed relatively mobile intermolecular characteristics, even though the dimer configuration

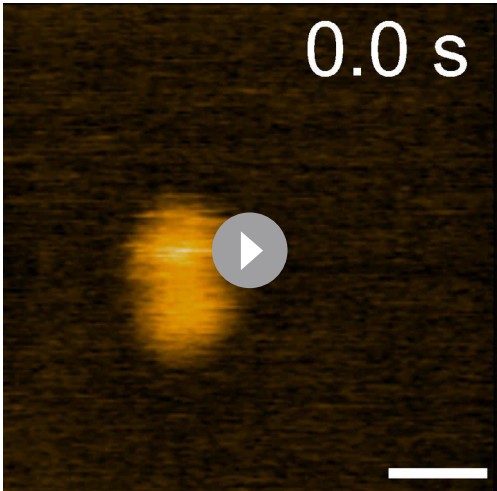

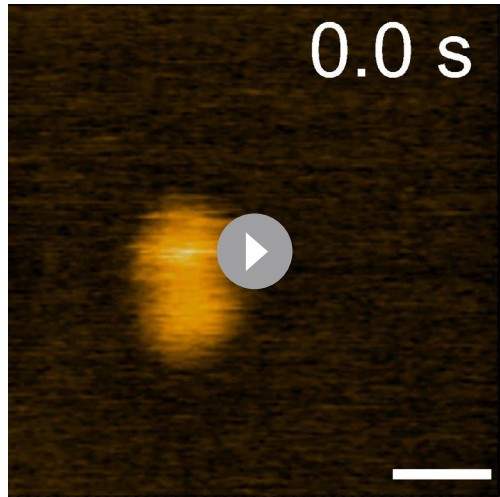

**Video 1.** AFM image of CDV-H. CDV_H, 100 × 100 nm. 5 fps, 5 frames per second, 100 × 100 nm scale, 200 × 200 pixel, scale bar: 20 nm.
https://elifesciences.org/articles/88929/figures#video1

**Video 2.** AFM image of CDV-H. CDV_H, 100 × 100 nm. 10 fps, 10 frames per second, 100 × 100 nm scale, 200 × 200 pixel, scale bar: 20 nm.
https://elifesciences.org/articles/88929/figures#video2

is dominant, which is in contrast with the static homodimer structures that are stabilized by several conserved interactions as observed in the crystals. In our previous report, immunoglobulins were observed by HS-AFM in essentially the same tapping mode conditions, clearly indicating their dynamical and structural features without any dissociation of their domains (*Yamazaki et al., 2020*). On the other hand, the observation of mobility of the receptor-binding domain (RBD) of SARS-CoV-2 S protein by HS-AFM was recently reported, showing that it forms open and closed conformation (*Lim et al., 2021*). Therefore, these results supported that the mobility of CDV-H head domains reflected its dynamic features without any artificial forces. The ability of the dimeric head domain to dissociate has already been shown in the NiV-G protein structure (*Wang et al., 2022*).

The previous model for morbillivirus entry only focused on the orientation shift of the dimers (*Hashiguchi et al., 2011*) or Cys154-centered pivot in MeV-H (*Navaratnarajah et al., 2011*) (alternative Gly158 hinge of CDV-H *Kalbermatter et al., 2023*); however, our results may suggest that further dissociation of the head domain homodimer via the connector region induced by receptor binding is relevant (*Figure 8*). In the proposed model, the dimers of MeV-H homodimers (tetramers) have stalk regions (and bottom faces), which are likely to attach to fusion proteins. Each H protein dimer is bridged by disulfide bond at 'neck' region located in top of stalk region. Step 1: The receptor (SLAM or Nectin-4) binds to the exposed region of head domains. Step 2: Receptor binding induces the dissociation of H tetramers into dimers. Step 3: The pulling of the receptor causes the head domains to further dissociate into individual domains (monomers). Step 4: Dislocation of individual H head domains influences the structural rearrangement of stalk regions, which allows fusion proteins to expose fusion peptides to access the membranes of target cells. Step 5: Fusion proteins undergo further structural changes to fuse the viral and cellular membranes to complete the fusion. In fact, Navaratnarajah conducted the mutagenesis study at the homodimer interface of MeV-H, revealing that the

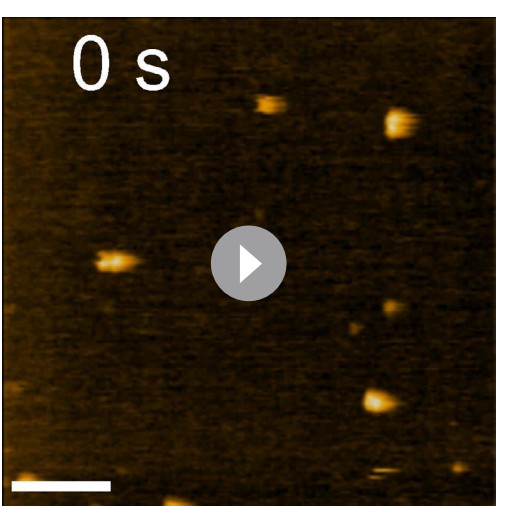

**Video 3.** AFM image of CDV-H. CDV_H, 500 × 500 nm. 5 fps, 5 frames per second, 500 × 500 nm scale, 200 × 200 pixel, scale bar: 100 nm.
https://elifesciences.org/articles/88929/figures#video3

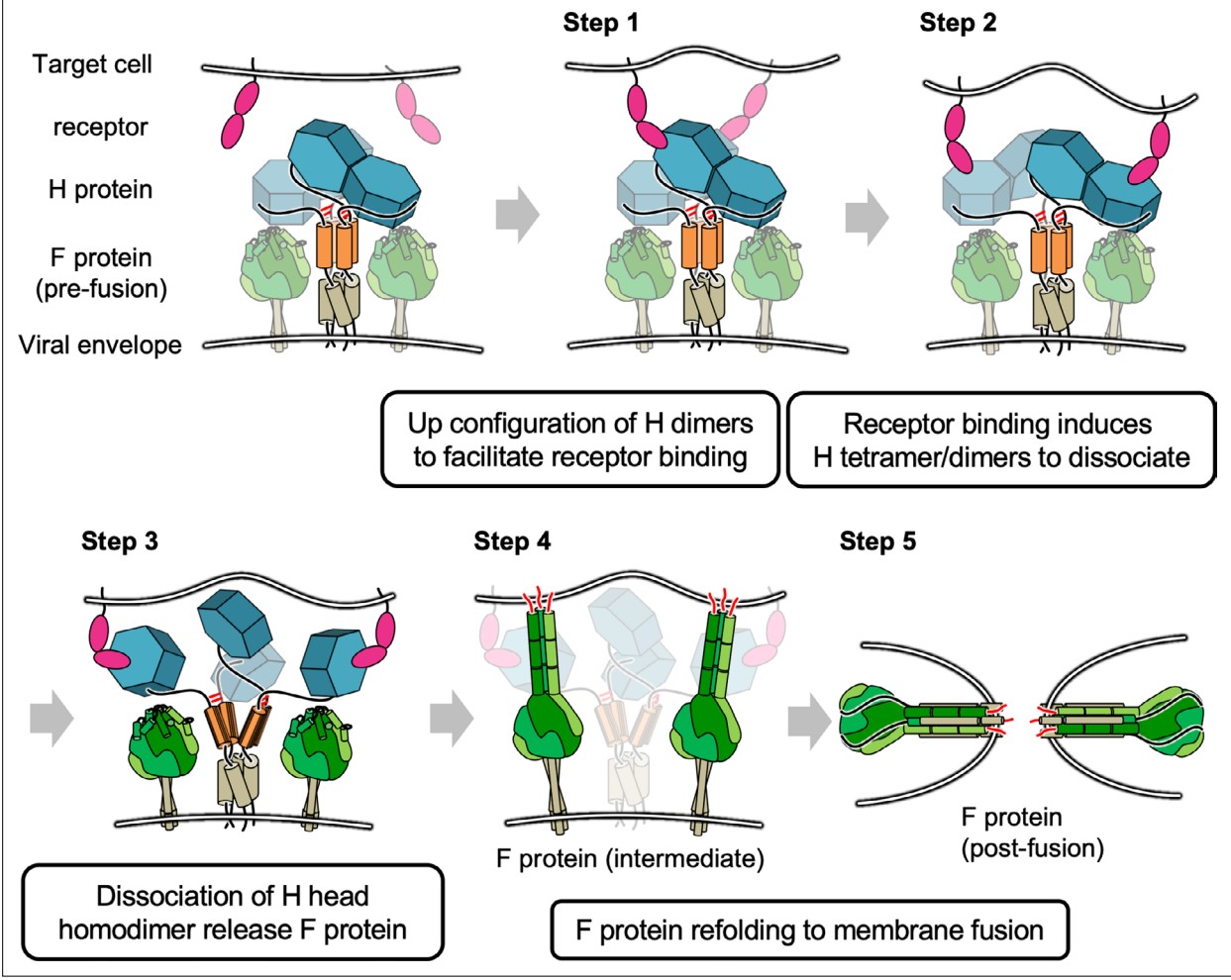

**Figure 8.** Proposed model of membrane fusion at canine distemper virus (CDV) entry. The homodimer dimers have stalk regions (colored orange) attached to fusion proteins. Each H protein dimer is bridged by disulfide bond (shown in red bar) at 'neck' region. Step 1: Binding of receptor (signaling lymphocyte activation molecule [SLAM] or Nectin-4) to the head domains. Steps 2 and 3: Receptor pulling causes the head domains to dissociate a dimer of homodimers and completely collapse to individual domains. Steps 4 and 5: Fusion proteins are allowed to access the membrane of target cells and complete fusion.

introductions of intermolecular disulfide bonds reduced the membrane fusion activity (*Navaratna-rajah et al., 2011*). A recent cryo-EM structure of the wild-type CDV-H ectodomain revealed that the head dimer is located on one side of the stalk region in solution (*Kalbermatter et al., 2023*), whereas cryo-EM analysis of human parainfluenza virus 3 showed that, on the viral envelope, the hemagglu-tinin–neuraminidase head interacts with the F protein in the up state (*Navaratnarajah et al., 2011*). Further biochemical and virological experiments are necessary to confirm the feasibility of these models. In addition, the closed areas at the interface of the homodimer could be exposed and thus have the potential to become epitopes targeted for anti-morbillivirus H neutralizing antibodies. These results provide significant insights into the viral fusion mechanism as well as the rational design of antiviral drugs and vaccines for a wide range of infectious diseases.

## Materials and methods
### Construction of expression plasmids
The expression plasmids encoding the ectodomain (amino acid residues Asn149 to Pro607 (or to Arg604)) of CDV-H derived from a Kyoto Biken vaccine strain (or a 5VD wild-type strain) were previously reported (*Fukuhara et al., 2019*). The cDNAs encoding the authentic signal sequence and ectodomain of human and dog Nectin-4 were amplified using the primer sets of 5′-GGCAAAGAAT

<u>TC</u>GCCACCATGCCCCTGTCCCTGGGAGCCG-3′ and 5′-CAGAAC<u>CTCGAG</u>AACATCCACAGTGACC TG-3′. The DNA fragments were cloned into a modified pCA7 vector using the restriction enzyme sites *Eco*RI and *Xho*I to fuse a C-terminal HRV 3C protease site, FLAG tag, His$_6$ tag, and Avi-tag sequence.

## Protein expression and purification

HEK293S cells lacking GnTI activity (*Aricescu et al., 2006*; *Reeves et al., 2002*) were kindly provided by Prof. Simon Davis (University of Oxford) and maintained in Dulbecco's modified Eagle medium (DMEM; Wako) supplemented with 10% fetal bovine serum (FBS; Hyclone) at 37°C and 5% $CO_2$. Eighty-% confluent cells were transfected with plasmids encoding CDV-H or Nectin-4 using poly-ethylenimine (Polyscience Inc) at a ratio of 1:2. After transfection, the cells were cultured in DMEM containing 2% FBS for 4 days. The soluble proteins secreted into the culture medium were purified by Ni$^{2+}$-NTA affinity chromatography (GE Healthcare) and Superdex 200 GL 10/300 gel filtration chroma-tography (GE Healthcare). The CDV-H protein for crystallization was further purified using Resource Q anion exchange chromatography (GE Healthcare).

## Western blot analysis

To determine the glycosylation sites on CDV-H, we disrupted the *N*-glycosylation potential sites by changing Asn to Asp (309, 391, 422, 456, 587, and 603) or deleting Asn149 of the CDV-H globular head domain. After they were subjected to SDS–PAGE and transferred to polyvinylidene difluoride (PVDF) membranes, anti-His-tag antibodies and horseradish peroxidase-conjugated goat anti-mouse IgG were added. Horseradish peroxidase activity was detected using Amersham ECL Plus Western Blotting Detection Reagent (GE Healthcare).

## Crystallization and structure determination

The CDV-H protein (4 mg/ml in 20 mM Tris–HCl pH 8.0, 100 mM NaCl) was crystallized by sitting-drop vapor diffusion at 20°C in 100 mM tri-sodium citrate pH 5.6, 10% polyethylene glycol-4000, and 10% isopropanol in 0.4 µl drops containing a 1:1 mixture of the protein to the reservoir solution. X-ray diffraction datasets were collected on NW-12 beamline at the Photon Factory (Tsukuba) and BL-41XU at Spring-8 (Harima) and integrated and scaled using HKL2000 (HKL Research, Inc) (*Minor et al., 2006*). The molecular replacement software, MOLREP, using the crystal structure of MeV-H (PDB ID code: 2Z6B) as a search model, provided a clear solution (*Hashiguchi et al., 2007*; *Vagin and Teplyakov, 2010*). Model refinement calculations were executed using REFMAC (*Murshudov et al., 2011*), and model building was performed using COOT (*Emsley and Cowtan, 2004*). Figures were prepared using PyMOL *Delano, 2002*.

## Surface plasmon resonance

Purified proteins were dissolved in N-(2-Hydroxyethyl)piperazine-N'-2-ethanesulfonic acid (HEPES) buffered saline (HBS) supplemented with ethylenediaminetetraacetic acid (EDTA) and surfactant P20 (HBS-EP) buffer (10 mM HEPES pH 7.4, 150 mM NaCl, 3 mM EDTA, and 0.005% Surfactant P20) (BIAcore AB). SPR experiments were performed using a BIACORE3000 instrument (GE Health-care). CDV-H was immobilized on a CM5 sensor chip (GE Healthcare) using the direct amine coupling method. dSLAM in HBS-EP buffer was injected over the immobilized CDV-H. Values obtained by subtracting the response in the negative controlled flow cell, which was immobilized bovine serum albumin from the response in sample flow cells, were regarded as the binding responses. The data were analyzed using the BIA evaluation version 4.1 (GE Healthcare) and ORIGIN version 7 software (Microcal Inc). The dissociation constant ($K_d$) was derived by the nonlinear curve fitting of the standard Langmuir binding isotherm. Kinetics data were fitted into rate equations derived from the simple 1:1 Langmuir binding model (A + B ↔ AB) using the curve-fitting model of BIA evaluation version 4.1.

## High-speed atomic force microscopy

HS-AFM images of CDV-H were acquired in the tapping mode at room temperature using an HS-AFM instrument, NanoExplorer (Research Institute of Biomolecule Metrology). The sample solution (2 µl) was placed on the freshly cleaved mica surface and incubated for 5 min. The samples on the mica were scanned in phosphate buffer (5 mM phosphate pH 6.2) by HS-AFM using a cantilever BL-AC10DS-A2 (Olympus). Images of 200 × 200 pixels from a 100 × 100 nm area were obtained at a scan rate of two

frames per second (fps). Images were analyzed using ImageJ v1.52p software (http://rsbweb.nih.gov/ij/; *Schneider et al., 2012*).

## Acknowledgements

We thank the beamline staff of the Photon Factory (Tsukuba, Japan) and SPring8 (Hyogo, Japan) for their technical help during data collection. We also thank Y Yanagi, I Tanaka, K Inaba, K Mihara, T Oka, H Aramaki, K Tokunaga, M., and K Sasaki for their discussions. This work was supported in part by the Japan Society for the Promotion of Science (JSPS) Grants-in-Aid for Scientific Research KAKENHI (Grants 15H02384, 20H03497 (to TH), and 22121007), the Scientific Research on Innovative Areas and International Group from the MEXT/JSPS KAKENHI [JP20H05873 (K Maenaka)], JSPS Strategic Young Researcher Overseas Visits Program for Accelerating Brain Circulation, the Japan Agency for Medical Research and Development (AMED) [JP223fa627005 (K Maenaka), JP20ae0101047, JP21fk0108463, JP21am0101093, JP22ama121037, JP24wm0325063 (K Maenaka), JP 22wm0325002h (to TH)], the Ministry of Health, Labour and Welfare of Japan, Hokkaido University Biosurface project, and Takeda Science Foundation.

## Additional information

### Funding

| Funder | Grant reference number | Author |
|---|---|---|
| Japan Society for the Promotion of Science | JP20H05873 | Katsumi Maenaka |
| Japan Society for the Promotion of Science | 22121007 | Katsumi Maenaka |
| Japan Society for the Promotion of Science | 15H02384 | Takao Hashiguchi |
| Japan Society for the Promotion of Science | 20H03497 | Takao Hashiguchi |
| Japan Agency for Medical Research and Development | JP223fa627005 | Katsumi Maenaka |
| Japan Agency for Medical Research and Development | JP20ae0101047 | Katsumi Maenaka |
| Japan Agency for Medical Research and Development | JP21fk0108463 | Katsumi Maenaka |
| Japan Agency for Medical Research and Development | JP21am0101093 | Katsumi Maenaka |
| Japan Agency for Medical Research and Development | JP22ama121037 | Katsumi Maenaka |
| Japan Agency for Medical Research and Development | JP24wm0325063 | Katsumi Maenaka |
| Japan Agency for Medical Research and Development | JP 22wm0325002h | Takao Hashiguchi |
| Hokkaido University | Biosurface project | Katsumi Maenaka |
| Takeda Science Foundation | Research project | Katsumi Maenaka |

| Funder | Grant reference number | Author |
|---|---|---|
| Japan Society for the Promotion of Science | JSPS Strategic Young Researcher Overseas Visits Program for Accelerating Brain Circulation | Katsumi Maenaka |

The funders had no role in study design, data collection, and interpretation, or the decision to submit the work for publication.

## Author contributions

Hideo Fukuhara, Conceptualization, Investigation, Writing - original draft, Writing - review and editing; Kohei Yumoto, Miyuki Sako, Investigation, Writing - original draft; Mizuho Kajikawa, Mihiro Kawamura, Mei Yoda, Surui Chen, Yuri Ito, Shin Takeda, Mwila Mwaba, Jiaqi Wang, Jun Kamishikiryo, Nobuo Maita, Chihiro Kitatsuji, Investigation; Toyoyuki Ose, Investigation, Writing - review and editing; Takao Hashiguchi, Makoto Takeda, Resources; Kimiko Kuroki, Supervision; Katsumi Maenaka, Conceptualization, Supervision, Funding acquisition, Investigation, Writing - original draft, Project administration, Writing - review and editing

## Author ORCIDs

Hideo Fukuhara http://orcid.org/0000-0002-7035-8206
Katsumi Maenaka https://orcid.org/0000-0002-5459-521X

Reviewer #2 (Public Review): https://doi.org/10.7554/eLife.88929.3.sa1
Author response https://doi.org/10.7554/eLife.88929.3.sa2

# Additional files

## Supplementary files

- MDAR checklist

## Data availability

Coordinates and structural factors were deposited in the Protein Data Bank under accession code 8YBF. Methods describing structure determination, binding analysis, atomic force microscopy, and assays for MeV entry and infection are described in the Materials and methods.

The following dataset was generated:

| Author(s) | Year | Dataset title | Dataset URL | Database and Identifier |
|---|---|---|---|---|
| Fukuhara H, Yumoto K, Sako M, Kajikawa M, Ose T, Hashiguchi T, Kamishikiryo J, Maita N, Kuroki K, Maenaka K | 2024 | Crystal structure of canine distemper virus hemagglutinin | https://www.rcsb.org/structure/8YBF | RCSB Protein Data Bank, 8YBF |

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
