## [Editor Report · eLife assessment]

The manuscript presents **valuable** findings, using **solid** techniques and approaches, that shed additional light into how the canine distemper virus (CDV) hemagglutinin might engage cellular receptors and how that engagement impacts host tropism. The structural data and their analysis were thorough and well-presented. The HS-AFM data, which indicate that homodimers may dissociate into monomers - and thus have significant implications for the model of fusion triggering - are very exciting, but require further validation, perhaps by alternate approaches, to bolster the current molecular model of the CDV fusion triggering.

---

## [Referee Report · Reviewer #2 (Public Review)]

The authors solved the crystal structure of CDV H-protein head domain at 3,2 A resolution to better understand the detailed mechanism of membrane fusion triggering. The structure clearly showed that the orientation of the H monomers in the homodimer was similar to that of measles virus H and different from other paramyxoviruses. The authors used the available co-crystal strictures of the closely related measles virus H structures with the SLAM and Nectin4 receptors to map the receptor binding site on CDV H. The authors also confirmed which N-linked sites were glycosylated in the CDV H protein and showed that both wildtype and vaccine strains of CDV H have the same glycosylation pattern. The authors documented that the glycans cover a vast majority of the H surface while leaving the receptor binding site exposed, which may in part explain the long-term success of measles virus and CDV vaccines. Finally, the authors used HS-AFM to visualize the real-time dynamic characteristics of CDV-H under physiological conditions. This analysis indicated that homodimers may dissociate into monomers, which has implications for the model of fusion triggering.

The structural data and analysis were thorough and well-presented. The HS-AFM data, while very exciting, needs to be further validated, perhaps by alternate approaches to further support the authors' model describing the molecular dynamics of fusion triggering.

---

## [Author Response]

The following is the authors’ response to the original reviews.

**eLife assessment**
Both reviewers positively received the manuscript, in general. The agreement was that the manuscript presented valuable findings, using solid techniques and approaches, that shed additional light into how the canine distemper virus hemagglutinin might engage cellular receptors and how that engagement impacts host tropism. While both reviewers appreciated the X-ray crystallographic data, they also felt that the AFM experiments could have been performed at a higher standard and that the interpretation of the results ensuing from those AFM experiments could have been explained more thoroughly and in simpler terms. An additional missed opportunity of the current manuscript is the lack of comparison of the crystal structure to that of the already published cryo-EM structure, for context.

Thank you very much for constructive comments of the editor and reviewers. Following your comments, we have changed the text related to the AFM experiments with simpler terms as follows.

“When CDV-H was loaded onto a mica substrate and scanned with a cantilever to acquire images of attached molecules, the CDV-H dimer was observed as two globules clustered together in most cases, but sometimes, each domain moved independently (Fig. 7B and Supplementary Movie). Time-course analysis of the dynamics of the representative CDV-H dimer showed that CDV-H could adopt both associated and dissociated forms (Fig. 7C). The distances between the domains were calculated by measuring those between the centers of mass of each domain. Finally, the distribution of distances between each head domain in the CDV-H dimers showed approximately 15 nm as a major peak (Fig. 7D). This is a reasonable length for the linker between the head domain dimers.” in Page 11, Lines 8-17.

With regards to the structural comparison between cryo-EM structure published in Proc. Natl. Acad. Sci. U. S. A. (2023) 120, e2208866120 and our crystal structure, we have compared these structures for Cα on page 6 and added the following text. “A recent cryo-EM structure of the wild-type CDV-H ectodomain revealed that the head dimer is located on one side of the stalk region in solution (Proc. Natl. Acad. Sci. U. S. A. (2023) 120, e2208866120)” in Page 14, Lines 22-24.

**Public Reviews:**

**Reviewer #1 (Public Review):**
Summary:Fukuhara, Maenaka, and colleagues report a crystal structure of the canine distemper virus (CDV) attachment hemagglutinin protein globular domain. The structure shows a dimeric organization of the viral protein and describes the detailed amino-acid side chain interactions between the two protomers. The authors also use their best judgement to comment on predicted sites for the two cellular receptors - Nectin-4 and SLAM - and thus speculate on the CDV host tropism. A complementary AFM study suggests a breathing movement at the hemagglutinin dimer interface.Strengths:The study of CDV and related Paramyxoviruses is significant for human/animal health and is very timely. The crystallographic data seem to be of good quality.

Thank you very much for the constructive comment of the reviewer.

Weaknesses:While the recent CDV hemagglutinin cryo-EM structure is mentioned, it is not compared to the present crystal structure, and thus the context of the present study is poorly justified. Additionally, the results of the AFM experiment are not unexpected. Indeed, other paramyxoviral RBP/G proteins also show movement at the protomer interface.

Thank you very much for constructive comments of the reviewer. When we submitted our manuscript to e-life, cryo-EM structure just published in Proc. Natl. Acad. Sci. U. S. A. (2023) 120, e2208866120 a week ago was not able to be available. Following the comment of the reviewer, we have added the text about the structural comparison between the cryo-EM structure and our crystal structure. We also have changed the text related to the AFM experiments to tone down the movement of the protomer interfaceas follows.

“This observation raises the possibility that each head domain of CDV-H also dissociates and moves flexibly, as shown in the structure of Nipah virus (NiV)-G protein, previously (Science (2022) 375, 1373–1378).” in Page 11, Lines 4-6.

**Reviewer #2 (Public Review):**
Summary:The authors solved the crystal structure of CDV H-protein head domain at 3,2 A resolution to better understand the detailed mechanism of membrane fusion triggering. The structure clearly showed that the orientation of the H monomers in the homodimer was similar to that of measles virus H and different from other paramyxoviruses. The authors used the available co-crystal strictures of the closely related measles virus H structures with the SLAM and Nectin4 receptors to map the receptor binding site on CDV H. The authors also confirmed which N-linked sites were glycosylated in the CDV H protein and showed that both wildtype and vaccine strains of CDV H have the same glycosylation pattern. The authors documented that the glycans cover a vast majority of the H surface while leaving the receptor binding site exposed, which may in part explain the long-term success of measles virus and CDV vaccines. Finally, the authors used HS-AFM to visualize the real-time dynamic characteristics of CDV-H under physiological conditions. This analysis indicated that homodimers may dissociate into monomers, which has implications for the model of fusion triggering.The structural data and analysis were thorough and well-presented. However, the HS-AFM data, while very exciting, was not presented in a manner that could be easily grasped by readers of this manuscript. I have some suggestions for improvement.(1) The authors claim their structure is very similar to the recently published croy-EM structure of CDV H. Can the authors provide us with a quantitative assessment of this statement?

Thank you very much for constructive comments of the reviewer. When we submitted our manuscript to e-life, cryo-EM structure just published in Proc. Natl. Acad. Sci. U. S. A. (2023) 120, e2208866120 a week ago was not able to be available. Following the comment of the reviewer, we have added the text about the structural comparison between the cryo-EM structure and our crystal structure. We also have changed the text related to the AFM experiments to tone down the movement of the protomer interface as follows.

“This observation raises the possibility that each head domain of CDV-H also dissociates and moves flexibly, as shown in the structure of Nipah virus (NiV)-G protein, previously (Science (2022) 375, 1373–1378).” in Page 11, Lines 4-6.

(2) The results for the HS-AFM are difficult to follow and it is not clear how the authors came to their conclusions. Can the authors better explain this data and justify their conclusions based on it?

Thank you very much for constructive comments of the reviewer. Following your comments, we have changed the text related to the AFM experiments with simpler terms as follows.

“When CDV-H was loaded onto a mica substrate and scanned with a cantilever to acquire images of attached molecules, the CDV-H dimer was observed as two globules clustered together in most cases, but sometimes, each domain moved independently (Fig. 7B and Supplementary Movie). Time-course analysis of the dynamics of the representative CDV-H dimer showed that CDV-H could adopt both associated and dissociated forms (Fig. 7C). The distances between the domains were calculated by measuring those between the centers of mass of each domain. Finally, the distribution of distances between each head domain in the CDV-H dimers showed approximately 15 nm as a major peak (Fig. 7D). This is a reasonable length for the linker between the head domain dimers.” in Page 11, Lines 8-17.

(3) The fusion triggering model in Figure 8 is ambiguous as to when H-F interactions are occurring and when they may be disrupted. The authors should clarify this point in their model.

Thank you very much for constructive comments of the reviewer. Following your comments, we have changed the Figure 8 and its legend.

**Recommendations for the authors:**

**Reviewer #1 (Recommendations For The Authors):**
(1) AFM experiments with SLAM or Nectin-4 immobilized on the cantilever would be much more informative.

Thank you very much for the constructive comment of the reviewer. We will try this experiment in the next paper.

(2) The authors should compare their crystal structure to that of the reported cryo-EM structure.

With regards to the structural comparison between cryo-EM structure published in Proc. Natl. Acad. Sci. U. S. A. (2023) 120, e2208866120 and our crystal structure, we have added the text.

(3) Figure 1D - why does the beta2 MG negative control have such a high SPR signal?

Thank you very much for the constructive comment of the reviewer. The immobilization levels for b 2-microglobulin (beta2 MG), CDV-OP-H and CDV-5VD-H were similar, 1204.7 RU, 1235.7 RU, and 1504.5 RU, respectively. We applied relatively high concentrations (5 mM) of dNectin4 and hNectin4 onto the chip to determine low-affinity dissociation constants. Then, the signals for beta2 MG (negative control) were high. In other SPR experiments for cell surface receptors, such high signals for beta2 MG were often observed in our previous paper, Kuroki et al., J. Immunol. 2019 Dec 15;203(12):3386-3394. doi: 10.4049/jimmunol.1900562. Therefore, we think that these SPR signals are not unusual.

(4) Figure 1C - please indicate the Ve volume for the peak and add in Ve for standard.

Thank you very much for the constructive comment of the reviewer. We have indicated the Ve volume for the peak and added in Ve for standard in Figure 1C.

(5) The authors mention that one of the chains in the asymmetric unit was better resolved than the other. Please show regions of the atomic model fit regions of the electron density to convince the reader of the quality of your data.

Thank you very much for the constructive comment of the reviewer. We have added new Supplementary figure 2 for comparison of electron density maps of chains A and B.

(6) Table 2 indicates that the difference between Rw and Rf values is larger than 5% which indicates slight overfitting during refinement. Please provide details of your refinement strategy and attempt simulated annealing as a strategy to reduce this delta.

Thank you very much for the constructive comment of the reviewer. We further introduced TLS and NCS parameters for the refinement. Consequently, the R/Rfree factors became 0.2645/0.3092. Simulated annealing had been already carried out. All the refinement statistics in the table 2 are updated.

**Reviewer #2 (Recommendations For The Authors):**
(1) The authors' fusion triggering model was difficult to follow. For example, this sentence was difficult to understand: "The other possible models may include the monomer-dimer-tetramer transition facilitated by receptor binding for the fusion."

Thank you very much for the constructive comment of the reviewer. Following your comments, we have removed the above sentences and have added the detail mechanism of the proposed model in Discussion. Furthermore, we have changed the Figure 8 and its legend for readers to understand more clearly.

(2) Figure 5A is not called out in the main text.

Thank you very much for the constructive comment of the reviewer. Following your comments, we have added the text as follows.

“the crystal structure of MeV-H in complex with hNectin-4 showed that the H-SLAM interaction consists of three main sites (Fig. 5A) (Nat. Struct. Mol. Biol. (2013) 20, 67–72).” in Page 11, Lines 4-6.

(3) Page 9, Line 4: interspaces? Perhaps interphases.

Thank you very much for the constructive comment of the reviewer. We have changed the term “interspaces” to “internal spaces”.

(4) Page 12, penultimate line: The authors mention "epitopes for anti-MeV-H Abs." Do they mean anti-CDV-H Abs?

Thank you very much for the constructive comment of the reviewer. Following your comments, we have changed the “anti-MeV-H Abs” to “anti-morbillivirus H neutralizing antibodies”.

(5) The paper will benefit from an English language editor to help clarify what the authors are trying to convey.

Thank you very much for the constructive comment of the reviewer.

We have asked a English proof reading company to check.